# Force plate methodologies applied to injury profiling and rehabilitation in sport: A scoping review protocol

**Francisco Javier Robles-Palazón**[1,2], **Paul Comfort**[1,3], **Nicholas J. Ripley**[1], **Lee Herrington**[1], **Christopher Bramah**[1], **John J. McMahon**[1] *

**1** Centre for Human Movement and Rehabilitation, University of Salford, Salford, United Kingdom, **2** Faculty of Sport Sciences, Department of Physical Activity and Sport, Campus of Excellence Mare Nostrum, University of Murcia, Murcia, Spain, **3** Centre for Exercise and Sport Science Research, Edith Cowan University, Joondalup, WA, Australia

* j.j.mcmahon@salford.ac.uk

## Abstract

Musculoskeletal injuries are a common health problem among sporting populations. Such injuries come with a high financial burden to the involved organisations and can have a detrimental impact on the career attainment of injured individuals. Force plates are now a common tool available to sport and exercise science and medicine professionals to enable them to profile injury risk predisposition and modulate the rehabilitation process within sporting environments. This is because contemporary force plate technology is portable and affordable and often comes with software that enables the automatic and immediate feedback of test variables to key stakeholders. However, to our knowledge, to date, there has been no comprehensive review of the scientific literature pertaining to clinical applications of force plate technology. Therefore, this article presents a protocol and a methodological framework to perform a scoping review to identify and map the available scientific literature in which force plates have been applied to the injury profiling and rehabilitation of athletes. The specific aims of the scoping review are 1) to identify and describe the force plate tests, methodologies, and metrics used to screen for injury risk and guide the return of injured athletes to full-time training and competition, 2) to identify potential trends and/or differences by participants' age, sex, and/or level of performance in tests, methodologies, and metrics selected, and 3) to identify key gaps in the existing evidence base and new questions that should be addressed in future research. The global aim of the scoping review is to improve practitioner decision-making around force plate test and variable selection when applied to the injury prevention and rehabilitation of sporting populations.

## 1. Introduction

Musculoskeletal injuries are a common health problem among sporting populations [1–4]. It has been proposed that a professional soccer team might expect around two musculoskeletal injuries per player every competitive season [5,6]. Likewise, three out of four elite athletes

**Data Availability Statement:** All relevant data from this study will be made available upon study completion.

**Funding:** This study was financially supported by the Spanish Ministry of Universities via the European Union–NextGenerationEU in the form of a Margarita Sala Postdoctoral Fellowship (UMU/R-1500/2021) awarded to FJR-P. No additional external funding was received for this study. The funder had no role in study design, data collection and analysis, decision to publish, or preparation of the manuscript.

**Competing interests:** The authors have read the journal's policy and have the following competing interests: PC and JJM supervise a PhD student who is not involved in this protocol or subsequent review but whose research is partly funded by Hawkin Dynamics LLC which is a force plate company. This does not alter our adherence to PLOS ONE policies on sharing data and materials. There are no patents, products in development or marketed products associated with this research to declare.

competing in athletics have reported at least one injury over a year of follow-up [4]. But the high incidence of injuries is not limited to adult professional sports. It has been estimated that around 40–60% of youth athletes participating in such popular team and individual sports as soccer, basketball, or athletics might suffer an injury over a typical competitive season as well [4,7,8]. As a result, musculoskeletal injuries are having an increased impact on sport participation, and may also require long rehabilitation processes [2]. The periods of absence may compromise athletes' skill maintenance and acquisition, which may then negatively influence their future performance [9–11]. These injury incidents will also lead to significant financial costs to sport clubs and governments worldwide [10,12]. In fact, an average English Premier League team might lose approximately £45 million per season due to injury-related decrements in performance [12]. Therefore, there is a clear necessity to develop and implement measures aimed at identifying those at risk from and reducing the number and severity of musculoskeletal injuries derived from participation in sports.

The implementation of appropriate training programs has shown to decrease the rate of injuries and their recurrences in athletes [13,14]. To be highly effective, training program designs must be targeted to each athlete's individual needs through the identification of specific predisposing factors [15]. The use of objective screening tools that allow sport and medicine professionals to profile injury risk predisposition and modulate the rehabilitation process may be considered key to the design of tailored preventive measures. Toward this effort, devices such as force plates, with the capability to measure and provide feedback about an athlete's force production characteristics [16], maximal strength [17], balance [18], running [19], and jumping and landing forces [26], have been developed. These devices are becoming increasingly utilised in applied environments such as sports [20,21] due to the advent of affordable, commercially available force plate systems that have been validated against industry gold standard systems [22–24] and well-established criterion data analyses procedures [25]. No longer, therefore, are most force plate tests being conducted via laboratory-grade systems located within a traditional research environment (e.g., University laboratories). In fact, millions of force plate tests are being conducted by practitioners each year, with this number likely to rise thanks to the quickness, portability, and valuable information that the modern force plate systems can provide practitioners without the requirement for additional technology, such as motion capture systems.

However, the increasingly frequent use of force plates also presents a challenge. On the one hand, the possibility of this equipment to conduct a large variety of assessments has led to many tests being proposed; a recent review has identified up to nine different tests to assess injury risk via jump assessments conducted with force plates [26]. However, in this review only prospective studies were analysed, so even more tests would be expected when considering other study designs [26]. On the other hand, while the high sampling frequencies generated by force plates (typically 1000 Hz) have allowed force-time metrics to be potentially highly sensitive to changes in the neuromuscular status of an individual, they have also presented a large amount of data and number of force-time metrics to choose from [27]. For example, for a countermovement jump (considered one of the simplest jump tests and one that can be pertinent to injury screening and rehabilitation [28]), more than one hundred different variables can be obtained immediately via commercial force plate software [29]. Many automatically generated force plate variables are duplicative or similar and so cluster analyses have been applied in a few recent studies to help practitioners identify relevant countermovement jump variables in an injury screening context [30,31].

The multitude of force plate tests and metrics, and sporting populations, that have been studied to date have created controversy and inconsistency within the scientific literature that makes synthesis of the findings difficult for a practitioner. Therefore, the existence of a review

would help to improve practitioner decision-making around force plate test and variable selection in relation to injury prevention purposes. After a preliminary search, no published or in-progress scoping or systematic reviews were identified on this topic, so here we present a protocol for a scoping review where we will provide a descriptive overview of the currently utilised force plate methodologies with athletes. In this protocol, we pre-define the objectives, methods, and reporting of our upcoming scoping review.

## 1.1. Aim

To establish the current evidence base underpinning the use of force plates to assess injury risk and return to sport purposes in competitive athletes.

## 1.2. Objectives

- To identify and describe the force plate tests, methodologies, and metrics used to screen athletes for injury risk.

- To identify and describe the force plate tests, methodologies, and metrics used to guide the return of injured athletes to full-time training and competition.

- To identify potential trends and/or differences by participants' age, sex, and/or level of performance in tests, methodologies, and metrics selected.

- To identify key gaps in the existing evidence base and new questions that should be addressed in future research.

## 2. Methodology

The framework proposed by the Joanna Briggs Institute (JBI) will be followed to conduct this scoping review [32,33]. The Preferred Reporting Items for Systematic Reviews and Meta-Analyses Protocols (PRISMA-P) [34] statement has been used to develop and report this protocol (S1 File). The Preferred Reporting Items for Systematic Reviews and Meta-Analyses (PRISMA 2020) [35] along with the specific extension for Scoping Reviews (PRISMA-ScR) [36] will be used to guide the reporting of the full scoping review findings (S2 File).

 As proposed by the JBI framework, the inclusion criteria will follow the elements of population, concept, and context [33]. A brief overview of all the eligibility criteria with their rationale can be found in S3 File.

## 2.1. Inclusion criteria

 **2.1.1. Participants.** The participants of interest for this review will be athletes, competing in either individual or team sports. No limitation by participants' age, sex, and/or level of performance will be applied. Due to the different injury profiles and demands of sports for these populations [37,38], which will probably result in the use of force plates with different objectives, studies on recreational athletes and paralympic athletes will be excluded from this review. To consider a study sample as "recreational athletes", the classification framework proposed by Mckay et al. [39] will be used (i.e., tier 1: those who meet World Health Organization minimum activity guidelines and/or participate in multiple sports/forms of activity, but do not train and compete regularly).

 **2.1.2. Concept.** Force plate methodologies applied to assess the injury profiling or rehabilitation process of athletes will be reviewed. For the purpose of this review, an injury will be

defined as any musculoskeletal damage, involving bones, muscles, ligaments, tendons, joints and associated tissues [40], derived from sport participation. No exclusion will be made based on the type of injury (e.g., primary and secondary injuries), as long as the study population is athletes (i.e., not former athletes) and, in those cases where the participants are injured, they are undergoing rehabilitation at the time of data collection. Illnesses and other non-musculo-skeletal damages will also not be considered. Tests and metrics identified might require a single or multiple force plates to screen the athlete. Although studies integrating complementary technology (e.g., three-dimensional motion analysis systems) will not be excluded, only the metrics extracted exclusively from data collected via force plates will be recorded. Those metrics derived from the combination of data obtained through different technologies (e.g., joint moments) will be beyond the scope of this review. The tests will be grouped into generic (i.e., used for multiple injury diagnoses) or injury-specific (e.g., knee injuries), upper body or lower body, unilateral or bilateral, single joint or multiple joints, and dynamic or isometric assessments. All this information will help guide practitioners and clinicians when choosing the most appropriate test and methodology.

**2.1.3. Context.** The context of this scoping review will be settings where athletes undergo physical assessments to evaluate their health status and where force plate methodologies are used to screen participants for injury risk (pre-injury event) and/or status of the rehabilitation process (post-injury). Any test applied during primary, secondary and/or tertiary injury prevention strategies will be identified and recorded. It should be noted that, in studies focused on primary injury prevention, a measure of the injury occurrence (i.e., prevalence and/or incidence) in the assessed cohort should be reported among the primary outcomes. Thus, those studies assessing the effect of an intervention (e.g., FIFA 11+) on athletes' physical performance measures (e.g., strength, balance) as well as those focused on proxy indicators of injury risk (e.g., knee valgus), with no report of injuries, will be excluded. As previously mentioned, studies involving injured athletes will be considered as long as the athletes are still symptomatic and undergoing their rehabilitation process; those involving athletes who have already completed the post-injury return-to-sport rehabilitation will then be excluded. Since the objective is to report on the methodologies used to guide practitioners' decision making based on the physical condition of athletes, studies evaluating the effect of protective equipment (e.g., knee brace, ankle taping) will not be considered either.

## 2.2. Sources

This review will consider any type of quantitative study designs, including randomised controlled trials, non-randomised controlled trials, quasi-experimental, before and after studies, prospective and retrospective cohort studies, case-control studies, and cross-sectional studies. Included studies will have to report their results in a paper published in a peer-reviewed journal. Literature reviews, conference abstracts, editorial commentaries, pre-prints, and letters to the editor will be excluded to avoid duplication of data.

## 2.3. Search strategy

A systematic search will be conducted in the databases MEDLINE/PubMed, Scopus, and Web of Science. In addition, a complementary search of the reference lists of included articles and a Google Scholar search will be also performed. This will be do through a backward (manually searching the reference list of a journal article) and forward (scanning a list of articles that had cited a given paper since it was published) citation tracking [41]. When additional studies that meet the inclusion criteria are identified, they will be included in the final pool of studies. Relevant search terms will be used to construct Boolean search strategies in the different databases.

**Table 1. Example of search strategy used in PubMed.**

| Source | Search strategy |
|--------|-----------------|
| PubMed | ("force plat*") AND (injur*[tiab] OR injur*[MeSH Terms] OR pain[tiab] OR pain[MeSH Terms] OR dysfunction[tiab] OR dysfunction[MeSH Terms] OR impairment[tiab] OR impairment[MeSH Terms] OR instability[tiab] OR instability[MeSH Terms] OR musculoskeletal[tiab] OR musculoskeletal[MeSH Terms] OR muscle[tiab] OR muscle[MeSH Terms] OR muscular[tiab] OR muscular[MeSH Terms] OR ligament[tiab] OR ligament[MeSH Terms] OR tendon[tiab] OR tendon[MeSH Terms] OR bone[tiab] OR bone[MeSH Terms] OR cartilage[tiab] OR cartilage[MeSH Terms] OR meniscus[tiab] OR meniscus [MeSH Terms]) AND (rehab*[tiab] OR rehab*[MeSH Terms] OR physio*[tiab] OR physio*[MeSH Terms] OR return*[tiab] OR return*[MeSH Terms] OR treatment*[tiab] OR treatment*[MeSH Terms] OR reconstruct*[tiab] OR reconstruct*[MeSH Terms] OR prevent*[tiab] OR prevent*[MeSH Terms] OR reduc*[tiab] OR reduc*[MeSH Terms] OR predict*[tiab] OR predict*[MeSH Terms] OR profil*[tiab] OR profil*[MeSH Terms] OR screen*[tiab] OR screen*[MeSH Terms] OR risk[tiab] OR risk[MeSH Terms]) AND (athlet*[tiab] OR athlet*[MeSH Terms] OR sport*[tiab] OR sport*[MeSH Terms] OR player*[tiab] OR player*[MeSH Terms]) |

An example of the initial search strategy that will be implemented in the MEDLINE/PubMed database can be found in Table 1. Studies published in English or Spanish will be considered for inclusion in this review. No date restrictions will be applied.

Two authors, independently, will carry out the selection of studies to be included in the scoping review [42]. To do this, a two-step search strategy will be used: first, studies will be screened based on title and abstract; second, the full text of the studies identified after the initial screening will be reviewed to identify those studies that meet all the inclusion criteria. A study will be excluded immediately when it failed to meet any of the eligibility criteria. Disagreements will be solved by consensus after consulting a third author. The entire study selection process will be carried out through Covidence (covidence.org), which will help to protect the integrity of the systematic review process and minimise bias [43]. The systematic search and study selection process will be regularly updated to identify new papers released during the development and writing of the review. To maximise the currency of our research, the last update will be carried out immediately after we have the first draft of the manuscript completed.

## 2.4. Extraction of results

A codebook will be designed to standardise the record of each study in order to maximise the objectivity of the data collection. The data collection form will be also pilot tested for usability and reliability by coding three to five randomly selected studies before the coding of the rest of the studies is started. Subsequently, each of the studies included in our review will be codified by two different authors. The moderator variables of the eligible studies will be coded and grouped into 3 categories: (1) general study descriptors (e.g., study design, year of publication, context, injury type/s); (2) study population (e.g., number of participants, age, sex, level of performance); and (3) force plate assessment characteristics. Force plate assessment characteristics reviewed will include test, instructions, number or time of registered trials, rest between trials, and metrics used. Other force plate characteristics, such as brand, model, sampling rate, filters, or software, will be also recorded. When necessary, the authors of the included studies will be contacted by email to provide clarifications. In any case, the codebook and data collection form might be adjusted during the review process if other potential variables of interest are identified. For example, additional characteristics considered worthy of study may be included during the review process, or variables that yield minimal information may be excluded. Data extraction will also be performed using the Covidence software.

## 2.5. Quality assessment

The methodological quality of the studies included in this review will be assessed in two ways: on the one hand, the reproducibility of the methodologies using force plates will be analysed by means of different variables included during the data extraction process (e.g., brand, model, sampling rate, filters applied); on the other hand, the quality of the research conducted will be assessed by means of a modified version of the Mixed-Methods Appraisal Tool (MMAT) 2018 [44] as well as by means of a modified version of the Oxford Centre for Evidence-Based Medicine (OCEBM) 2009 model [45]. The MMAT has been used for the assessment of the quality of studies included in previous scoping reviews, and its comprehensiveness, usefulness, and reliability has been widely demonstrated [46]. For the purposes of our scoping review, only those categories of the appraisal tool focused on quantitative studies (i.e., categories 2, 3, and 4) will be applied. The methodological quality for each included study will be described using the corresponding MMAT criteria, and where appropriate, an overall quality score will be calculated. The OCEBM will be used to determine level of evidence for each study. Based on the studies' exclusion criteria, the levels 1a, 2a, 3a (systematic reviews), and 5 (opinion-based papers) will be excluded in this case. The quality assessments will be also conducted by two independent authors, and any discrepancies during this process will be settled by consensus after consultation with a third author.

## 2.6. Presentation of results

All the tests identified will be summarised in a table presenting what (metrics), when (screening for injury risk or return to sport purposes), where (type of sport), how (range of sets, repetitions and/or time; metrics), and how much (frequency) are used in the literature reviewed. Additionally, a brief description of the tests and metrics will be provided. The information provided in this review should not only aid sport and exercise science and medicine researchers and practitioners in making informed decisions when applying force plate methods to the injury profiling and return to sport processes of athletes (sport or tactical), but also force plate manufacturers and technical developers in understanding which are the main applications of this equipment on a scientific and practical level to date.

## 3. Discussion

The high injury rates shown in male and female, youth and adult, and elite and sub-elite athletes [1–4,7,8] combined with the large opportunities offered by force plates for injury screening and rehabilitation in applied environments justify the development of this scoping review. The current protocol outlines the specific objectives, methodology, and reporting of the planned review, which will help increase transparency, and reduce duplication and publication bias [33]. Two researchers performing the entire process of search, selection, and extraction of data independently will also increase the accuracy of the data recorded and reduce errors associated with this process. Despite these strengths, we acknowledge that the restriction by language (including only studies written in English and Spanish) and publication type (only studies published in peer-reviewed journals) will be a limitation of our scoping review. However, we also consider that these restrictions will facilitate the completion of our work and thus enable the results to reach sport and exercise coaches and clinicians sooner, and they will not have a major impact on the conclusions drawn in our review.

## Supporting information

**S1 File. Preferred Reporting Items for Systematic reviews and Meta-Analyses Protocols (PRISMA-P) checklist.**
(DOC)

**S2 File. Preferred Reporting Items for Systematic reviews and Meta-Analyses extension for Scoping Reviews (PRISMA-ScR) checklist.**
(DOCX)

**S3 File. Inclusion/Exclusion criteria for literature search.**
(DOCX)

## Author Contributions

**Conceptualization:** John J. McMahon.

**Methodology:** Francisco Javier Robles-Palazón.

**Project administration:** John J. McMahon.

**Resources:** Francisco Javier Robles-Palazón.

**Supervision:** Paul Comfort.

**Writing – original draft:** Francisco Javier Robles-Palazón, Paul Comfort, Nicholas J. Ripley, Lee Herrington, Christopher Bramah, John J. McMahon.

**Writing – review & editing:** Francisco Javier Robles-Palazón, Paul Comfort, Nicholas J. Ripley, Lee Herrington, Christopher Bramah, John J. McMahon.

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
