## [Decision Letter · Decision Letter 0]

3 May 2023

PONE-D-23-07414Force plate methodologies applied to the injury profiling and rehabilitation of sport and tactical athletes: a scoping review protocolPLOS ONE

Dear Dr. McMahon,

Thank you for submitting your manuscript to PLOS ONE. After careful consideration, we feel that it has merit but does not fully meet PLOS ONE’s publication criteria as it currently stands. Therefore, we invite you to submit a revised version of the manuscript that addresses the points raised during the review process.

Both reviewers have recognized the quality of the written work submitted. However, both  have pointed out several major areas of concern regarding necessary methodological omissions, which if addressed/included would vastly improve the quality of the work presented. Justification is required for both the need for the scoping review and the protocol itself, highlighting their importance to the 'field', and adhering to reporting standards for review document/papers.

We look forward to receiving your revised manuscript.

Kind regards,

Chris Connaboy

Academic Editor

PLOS ONE

Journal Requirements:

"YES

FJRP was supported by a Margarita Salas postdoctoral fellowship (UMU/R-1500/2021) given by the Spanish Ministry of Universities and funded by the European Union–NextGenerationEU. The funders did not and will not have a role in study design, data collection and analysis, decision to publish, or preparation of the manuscript."

"I have read the journal's policy and the authors of this manuscript have the following competing interests: JJM and PC supervise a PhD student whose research which is jointly funded by the University of Salford and Hawkin Dynamics Inc. which is a force plate hardware and software manufacturer. The funders did not and will not have a role in study design, data collection and analysis, decision to publish, or preparation of the manuscript." 

6. We note that this manuscript is a systematic review or meta-analysis; our author guidelines therefore require that you use PRISMA guidance to help improve reporting quality of this type of study. Please upload copies of the completed PRISMA checklist as Supporting Information with a file name “PRISMA checklist”.

Reviewers' comments:

Reviewer's Responses to Questions

**Comments to the Author**

1. Does the manuscript provide a valid rationale for the proposed study, with clearly identified and justified research questions?

Reviewer #1: Yes

Reviewer #2: Partly

2. Is the protocol technically sound and planned in a manner that will lead to a meaningful outcome and allow testing the stated hypotheses?

Reviewer #1: Partly

Reviewer #2: Partly

3. Is the methodology feasible and described in sufficient detail to allow the work to be replicable?

Reviewer #1: Yes

Reviewer #2: Yes

4. Have the authors described where all data underlying the findings will be made available when the study is complete?

Reviewer #1: Yes

Reviewer #2: No

5. Is the manuscript presented in an intelligible fashion and written in standard English?

Reviewer #1: Yes

Reviewer #2: Yes

6. Review Comments to the Author

You may also provide optional suggestions and comments to authors that they might find helpful in planning their study.

Reviewer #1: Efforts to prevent musculoskeletal injuries and improve clinical recovery outcomes remain significant challenges for healthcare providers. More recently, portable force plates have provided new scientific capabilities to potentially identify athletes at risk for sustaining an injury or evaluate functional performance following recovery. Currently, as the authors describe, there has yet to be scoping review of the relevant literature describing the methodologies of force plates among both sport and tactical (law enforcement, first responder/paramedic, etc.) to be used for identifying both injury profiling (injury risk) and rehabilitation (recovery progression). Overall, the manuscript is well-written and the supporting documents (supplementary Tables 1-3) are correctly identified in the current submission. However, significant limitations in the methodological approach leave the scoping review too broad which would significantly reduce the overall scientific impact of resultant findings. The topic is appropriate for PLOS One but requires significant improvements before further consideration.

In short, the authors propose to review any peer reviewed article (written in English or Spanish) examining at least one force plate outcome among sport and tactical athletes that may be related to injury onset (risk of injury) or physical recovery (injury rehabilitation) by including prospective and retrospective cohort (including randomized and non-randomized), cross-sectional, and case-control studies. Because of including a broad population (sport and tactical athletes) and potential outcomes (any force pate measurement outcome) associated with multiple contexts (Injury profiling and injury rehabilitation), there is considerable likelihood of a high number of irrelevant articles to fulfill the aim of this study. Additionally, just because both sport (recreational and competitive) and tactical athletes experience musculoskeletal injuries during training and/or fulfilling job duties, the authors need to improve their justification why force plate measurement applications can equally apply to both populations. Injury is also not clearly defined in the protocol, which would greatly affect the potential number of studies and refine the focus of the current protocol. As currently written, potential articles would include studies measuring proxy indicators of injury risk (e.g., knee valgus during squat) without measuring injury occurrence itself (for most cross-sectional studies). If the current protocol plans to conduct a scoping review of this breadth, then the Methods section should clarify these considerations.

In summary, there is a need for a scoping review to aggregate the relevant literature examining force plate methodologies used for injury profiling and to inform injury recovery, but the protocol as currently written is too broad and needs further refinement to ensure that resultant articles clearly align with the study objectives.

Title:

The title is directional and clearly conveys the manuscript. No additional comment.

Abstract

The abstract is succinct and appropriately describes the study protocol.

Introduction

In the first paragraph of the Introduction, please provide examples of injury risk among tactical populations. Currently, the introduction pertains to athletes and U.S. army, can you provide examples for law enforcement, fire and rescue, paramedics, or EMTs? Since tactical populations are also a focal point of the scoping review, the rationale to support the need for these populations would be beneficial.

Line 76-78: Please provide a citation for each of the potential outcomes that have been developed from force plates, including “an athlete’s force production characteristics (CITE), maximal strength (CITE), balance (CITE), running (CITE), and jumping and landing forces (CITE).

Methods

2.1.1. Please clarify if indicators or markers of injury predisposition will be included (I assume the author team does not intend to include these studies). This can present many studies that screened healthy athletes that underwent force plate measurement that was not included with any injury.

2.1.1. Line 150: Will all participants derive from organized sporting groups, and not include recreational athletes? In line 50, ‘level of performance’ implies competitive levels (e.g., high school vs college or amateur vs advanced) within a particular organized sport. I am not certain if recreational athletes are intended in the current review. If so, this may present a limitation due to varying injury exposure levels among recreational athletes compared to those actively participating in a competitive season. Can you please clarify and provide the rationale whether you will include these recreational athletes?

For included articles, will data be extracted in duplicate, or will only 1 reviewer extract data and 2nd reviewer confirm that extracted information is correct?

Given the anticipated breadth of articles that will apply to the search criteria with minimal exclusionary criteria, it would be important to identify if (or when) an update to the literature search will occur. Please clarify your planned approach to address either 1 or multiple literature searches.

Line 203: I believe ‘any’ should be referred to ‘all’ eligibility criteria. Please review and correct if necessary.

Line 209: Among the categories, 1) general study descriptors, 2) study population, 3) force plate assessment characteristics, only force place characteristics are further clarified (e.g., instructions, number of trials, rest between trials, etc.). Can you please describe the general study descriptors and study population?

Line 221: Recommend rephrasing ‘where’ to ‘who’ when describing sport or tactical athletes.

Reviewer #2: The following manuscript proposes a comprehensive scoping review on force plate testing methodology and their use for injury risk identification, prevention, and rehabilitation. There is a need for this review as force plate use as a means to assess athletes (sporting and tactical) has become very popular and extends beyond the laboratory now. Additionally, the proliferation of commercial force plate companies has resulted in the over saturation of information regarding their use making it difficult for the practitioners to navigate the optimal use strategies. I commend the authors for taking on this topic as I think it is much needed in the sports medicine field. However, I have several concerns regarding this proposed protocol.

Firstly, the protocol appears to be a little too ambitious given the limited exclusion criteria. Including athletic and tactical populations and investigations of any study design will result in a large number of studies in the final report which is a lot to undertake for a single review. I would suggest delimiting to only athletics or tactical populations since there are valid arguments that these populations have very different mechanical exposures. Additionally, it is unclear why lower quality study designs would be included in this review. One of the current issues with this topic is the high number of cross-sectional studies that link a force plate assessment to injury prediction without direct evidence.

Secondly, I was surprised to see no methods listed to discern publication quality, risk of bias, etc. There are many validated tools available to objectively appraise the literature being reviewed and would help identify the state of the science regarding force plates and musculoskeletal injuries (objective 3). Furthermore, this would help identify limitations and gaps in the science that need to be addressed with future work (objective 4). Would suggest reviewing some protocols published in the journal ‘systematic reviews’ and the standards for a Cochrane review to get ideas to create a more novel and rigorous protocol for this scoping review.

Lastly, my biggest issue is it is not clear why this scoping review protocol should be published on its own separate from the actual scoping review itself. The introduction does a fine job justifying the need for this scoping review (which I agree a review on this topic is needed). However, from what is presented in the manuscript, this protocol is not using any novel methods to appraise the literature nor is it the most thorough protocol I’ve seen. The questions being asked are large in scope, but it is missing key rigors of objectivity that would set it apart (e.g., study quality assessment, risk of bias tools, etc). If there is a valid reason for the publishing of the protocol on its own, then it has not been adequately conveyed (this case needs to be made in the introduction).

Comments for specific sections below.

Abstract:

Line 33: Change ‘systematic review’ to just ‘review’ since you go on to propose a scoping review.

Line 50: Remove ‘systematic review’ from keywords since this is technically not a systematic review

Introduction:

While I think the rationale for why a scoping review is necessary is acceptable, the introduction provides no strong rationale as to why the literature needs a manuscript describing the methods of a proposed scoping review. What necessitates the need for knowing the methodological rigor (which would be described in the methods section of the scoping review anyways)? What is novel about your methodological approach to the scoping review? What gaps or limitations in the literature are these novel methods addressing? These are the types of questions that need to be addressed/discussed in the introduction for a protocol paper.

Specific comments below:

Line 72: phrase ‘allow sport and exercise science and medicine professionals’ is clunky, rephrase to improve readability.

Lines 78-90: Most of the information is redundant. Only need to discuss the low cost and accessibility of force plates one time.

Lines 108-116: This paragraph is somewhat misleading as it implies that a scoping review was performed, not the proposal of a methodological framework for a potential scoping review.

Aim/Objectives: These are very ambitious and will result in a very large number of papers for the final analysis (even after screening papers out)

Methodology:

2.1.1 Participants:

Why no age or level of performance exclusion criteria? You are including both sport and tactical populations, but excluding paralympic athletes due to ‘different injury profiles and demands of sports for this populations’; does that mean you consider injury profiles of traditional athletes and tactical athletes analogous? Do you think injury profiles, exposure and sports demands are similar between adolescent recreational athletes (e.g., high school varsity basketball) and even division I collegiate basketball players?

2.1.2 Concept:

Line 158: are you only considering primary injuries or secondary musculoskeletal injuries as well such as osteoarthritis?

2.2 Sources:

This is a very broad range of studies to include which will drastically increase the number of manuscripts you will be reviewing. Do you believe that quasi-experimental, case-control, and cross-sectional studies will provide valid evidence for a force plate and its associated test’s prognostic capability? I would argue it would not and will further muddy interpretations interfering with the objective to provide a use guide for practitioners.

2.3 Search strategy:

If going to include tactical athlete populations would suggest searching DTIC as well since much of their work won’t be published in journals indexed in Medline.

Between what years is the search being conducted?

Line 199-201: Need to have a specific statements or questions for both levels of screening that is reported to show how you identified if study was included or excluded. Would suggest using at minimum three levels (i.e., title only screening, abstract and full text).

Discussion:

Line 233-236: This is true that it will increase transparency, but all this information would be presented in the actual review as well.

Line 236: Only two individuals screening for a review this size might take a long time, especially during the stages of data extraction.

Line 240: I wouldn’t consider only reviewing peer-reviewed literature as a weakness for this review since there is also a lot of misinformation on force plate injury screening protocols (especially from commercial sector just trying to sell their force plate and software platform).

7. PLOS authors have the option to publish the peer review history of their article (what does this mean?). If published, this will include your full peer review and any attached files.

Reviewer #1: No

Reviewer #2: No

---

## [Author Response · Author response to Decision Letter 0]

19 Jul 2023

Dear editor and reviewers,

Thank you very much for your comments. We appreciate your valuable feedback and feel that your comments have contributed to an improved protocol. Furthermore, we have been forced to reflect and have learnt a lot from the reviewers´ suggestions and comments, which will definitely help to improve the quality of our scoping review as well. Below, we have addressed all your comments point by point in green. In addition, you can find the changes in the main manuscript highlighted with track changes.

Reviewer #1: 

Efforts to prevent musculoskeletal injuries and improve clinical recovery outcomes remain significant challenges for healthcare providers. More recently, portable force plates have provided new scientific capabilities to potentially identify athletes at risk for sustaining an injury or evaluate functional performance following recovery. Currently, as the authors describe, there has yet to be scoping review of the relevant literature describing the methodologies of force plates among both sport and tactical (law enforcement, first responder/paramedic, etc.) to be used for identifying both injury profiling (injury risk) and rehabilitation (recovery progression). Overall, the manuscript is well-written and the supporting documents (supplementary Tables 1-3) are correctly identified in the current submission. However, significant limitations in the methodological approach leave the scoping review too broad which would significantly reduce the overall scientific impact of resultant findings. The topic is appropriate for PLOS One but requires significant improvements before further consideration.

In short, the authors propose to review any peer reviewed article (written in English or Spanish) examining at least one force plate outcome among sport and tactical athletes that may be related to injury onset (risk of injury) or physical recovery (injury rehabilitation) by including prospective and retrospective cohort (including randomized and non-randomized), cross-sectional, and case-control studies. Because of including a broad population (sport and tactical athletes) and potential outcomes (any force pate measurement outcome) associated with multiple contexts (Injury profiling and injury rehabilitation), there is considerable likelihood of a high number of irrelevant articles to fulfill the aim of this study. Additionally, just because both sport (recreational and competitive) and tactical athletes experience musculoskeletal injuries during training and/or fulfilling job duties, the authors need to improve their justification why force plate measurement applications can equally apply to both populations. Injury is also not clearly defined in the protocol, which would greatly affect the potential number of studies and refine the focus of the current protocol. As currently written, potential articles would include studies measuring proxy indicators of injury risk (e.g., knee valgus during squat) without measuring injury occurrence itself (for most cross-sectional studies). If the current protocol plans to conduct a scoping review of this breadth, then the Methods section should clarify these considerations. In summary, there is a need for a scoping review to aggregate the relevant literature examining force plate methodologies used for injury profiling and to inform injury recovery, but the protocol as currently written is too broad and needs further refinement to ensure that resultant articles clearly align with the study objectives.

Thank you very much for reviewing our manuscript. We are really grateful for your work and agree with many of the points you made. In fact, considering the question raised about the breadth of the intended review, we have decided to focus our study population on athletes competing in individual and team sports exclusively (excluding tactical athletes and recreational athletes). In addition, we have clarified in the context of the study that only those papers that include injury occurrence outcomes and/or involve injured athletes at the time of the study will be considered. Thus, those investigations that analyse performance measures and/or proxy indicators of injury risk (e.g., knee valgus during squat) without measuring injury occurrence itself will be excluded. Thanks to these modifications, we believe that the review will provide more accurate and relevant results for practitioners, while reducing the timeframe for its development and dissemination of results.

Below, we respond in detail to the specific comments made for some of the sections of the protocol.

Title:

The title is directional and clearly conveys the manuscript. No additional comment.

Thank you for the comment. We have kept the same title, with the only modification being the deletion of the term “tactical” since the focus of the review has been changed to sports athletes.

Abstract

The abstract is succinct and appropriately describes the study protocol.

Thank you. As for the title, only those parts relating to tactical athletes have been removed.

Introduction

In the first paragraph of the Introduction, please provide examples of injury risk among tactical populations. Currently, the introduction pertains to athletes and U.S. army, can you provide examples for law enforcement, fire and rescue, paramedics, or EMTs? Since tactical populations are also a focal point of the scoping review, the rationale to support the need for these populations would be beneficial.

As tactical athletes are no longer a focal point of our scoping review, the information related to this population has been removed from this paragraph. Contrarily, we have included some examples of injury risk among different athletes, to highlight the importance of reducing them across diverse sports, ages and levels of play. The following are examples of some of the new sentences included:

Likewise, three out of four elite athletes competing in athletics have reported at least one injury over a year of follow-up (4). But the high incidence of injuries is not limited to adult professional sports. It has been estimated that around 40-60% of youth athletes participating in such popular team and individual sports as soccer, basketball, or athletics might suffer an injury over a typical competitive season as well (4,7,8).

Line 76-78: Please provide a citation for each of the potential outcomes that have been developed from force plates, including “an athlete’s force production characteristics (CITE), maximal strength (CITE), balance (CITE), running (CITE), and jumping and landing forces (CITE).

Thank you for this comment. We have added citations to support these common applications of force plates in line with your suggestion. 

Methods

2.1.1. Please clarify if indicators or markers of injury predisposition will be included (I assume the author team does not intend to include these studies). This can present many studies that screened healthy athletes that underwent force plate measurement that was not included with any injury.

Thank you for this comment. This has been clarified in the section “2.1.3. Context”.

2.1.1. Line 150: Will all participants derive from organized sporting groups, and not include recreational athletes? In line 50, ‘level of performance’ implies competitive levels (e.g., high school vs college or amateur vs advanced) within a particular organized sport. I am not certain if recreational athletes are intended in the current review. If so, this may present a limitation due to varying injury exposure levels among recreational athletes compared to those actively participating in a competitive season. Can you please clarify and provide the rationale whether you will include these recreational athletes?

Thank you for this comment. As the reviewer indicates, it would be very difficult to compare recreational vs. competitive athletes because of the differences in injury exposure levels so, in our review, we will only include athletes participating in an organised sport. Thus, recreational athletes (i.e., those who meet World Health Organization minimum activity guidelines and/or participate in multiple sports/forms of activity, but do not train and compete regularly [McKay et al., 2022]) will be excluded. This has been detailed in the main text of the protocol, and also in the Supplementary File 3 where inclusion and exclusion criteria are summarised.

For included articles, will data be extracted in duplicate, or will only 1 reviewer extract data and 2nd reviewer confirm that extracted information is correct?

Study characteristics and force plate outcome data will be coded in duplicate by two reviewers to reduce both the risk of making mistakes and the possibility that data selection/extraction is influenced by a single person’s biases (Peters et al., 2015; Peters et al., 2020). To maximise the objectivity of the coding, a codebook and a data collection form will be designed to standardise the record of each study. The data collection form will be pilot tested for usability and reliability by coding three to five randomly selected studies before the coding of the rest of the studies is started. In any case, unavoidable disagreements during the final codification process of included studies will be solved through consensus or by consulting a third author. This information has been added to the “2.4. Extraction of results” section of the protocol.

Given the anticipated breadth of articles that will apply to the search criteria with minimal exclusionary criteria, it would be important to identify if (or when) an update to the literature search will occur. Please clarify your planned approach to address either 1 or multiple literature searches.

As the reviewer indicates, the breadth of articles included for review in our research may delay the process of performing, writing, and publishing our scoping review. We are aware of that and, although we will make important efforts to have the final report as soon as possible, this is the reason why we have planned to regularly update the search, with the last update being just when we have the first draft of the scoping review manuscript completed. To do that, we will conduct all the study selection (and also data extraction) through Covidence (covidence.org), a software for managing systematic reviews which will de-duplicate files upon import new references. This will help us to identify new papers that have recently been published as well as to maximise the currency of our review. All this information has been added to the protocol as follows:

The entire study selection process will be carried out through Covidence (covidence.org), which will help to protect the integrity of the systematic review process and minimise bias (43). The systematic search and study selection process will be regularly updated to identify new papers released during the development and writing of the review. To maximise the currency of our research, the last update will be carried out immediately after we have the first draft of the manuscript completed.

Line 203: I believe ‘any’ should be referred to ‘all’ eligibility criteria. Please review and correct if necessary.

If a study does not meet one of the criteria, it will be excluded. This is why we think that “any” would be more appropriate here.

Line 209: Among the categories, 1) general study descriptors, 2) study population, 3) force plate assessment characteristics, only force place characteristics are further clarified (e.g., instructions, number of trials, rest between trials, etc.). Can you please describe the general study descriptors and study population?

We have included some examples for general study descriptors and study population characteristics as suggested.

Line 221: Recommend rephrasing ‘where’ to ‘who’ when describing sport or tactical athletes.

We have changed "athletes" to "type of sport", so we have kept "where".

References used in responses to Reviewer 1:

McKay AKA, Stellingwerff T, Smith ES, Martin DT, Mujika I, Goosey-Tolfrey VL, et al. Defining training and performance caliber: a participant classification framework. Int J Sports Physiol Perform. 2022;17(2):317–31.

Peters MDJ, Godfrey CM, Khalil H, McInerney P, Parker D, Soares CB. Guidance for conducting systematic scoping reviews. JBI Evid Implement. 2015;13(3):141–6. 

Peters MDJ, Godfrey CM, McInerney P, Munn Z, Tricco AC, Khalil H. Chapter 11: Scoping Reviews. In: Aromataris E, Munn Z, editors. JBI Manual for Evidence Synthesis. JBI; 2020.

Reviewer #2: 

The following manuscript proposes a comprehensive scoping review on force plate testing methodology and their use for injury risk identification, prevention, and rehabilitation. There is a need for this review as force plate use as a means to assess athletes (sporting and tactical) has become very popular and extends beyond the laboratory now. Additionally, the proliferation of commercial force plate companies has resulted in the over saturation of information regarding their use making it difficult for the practitioners to navigate the optimal use strategies. I commend the authors for taking on this topic as I think it is much needed in the sports medicine field. However, I have several concerns regarding this proposed protocol.

Firstly, the protocol appears to be a little too ambitious given the limited exclusion criteria. Including athletic and tactical populations and investigations of any study design will result in a large number of studies in the final report which is a lot to undertake for a single review. I would suggest delimiting to only athletics or tactical populations since there are valid arguments that these populations have very different mechanical exposures. Additionally, it is unclear why lower quality study designs would be included in this review. One of the current issues with this topic is the high number of cross-sectional studies that link a force plate assessment to injury prediction without direct evidence.

Secondly, I was surprised to see no methods listed to discern publication quality, risk of bias, etc. There are many validated tools available to objectively appraise the literature being reviewed and would help identify the state of the science regarding force plates and musculoskeletal injuries (objective 3). Furthermore, this would help identify limitations and gaps in the science that need to be addressed with future work (objective 4). Would suggest reviewing some protocols published in the journal ‘systematic reviews’ and the standards for a Cochrane review to get ideas to create a more novel and rigorous protocol for this scoping review.

Lastly, my biggest issue is it is not clear why this scoping review protocol should be published on its own separate from the actual scoping review itself. The introduction does a fine job justifying the need for this scoping review (which I agree a review on this topic is needed). However, from what is presented in the manuscript, this protocol is not using any novel methods to appraise the literature nor is it the most thorough protocol I’ve seen. The questions being asked are large in scope, but it is missing key rigors of objectivity that would set it apart (e.g., study quality assessment, risk of bias tools, etc). If there is a valid reason for the publishing of the protocol on its own, then it has not been adequately conveyed (this case needs to be made in the introduction).

Thank you very much for reviewing our manuscript. We are really grateful for your work and consider that some of your suggestions have made us improve the proposed protocol. In fact, in line with your first comment, we have limited the study population in our review to sport athletes only. Following your second comment, we have also selected two scales/methods to assess the quality of the studies included in our review. Finally, and with regard to your last comment, the publication of an a priori protocol has been described as an important step to increase the transparency and quality of the review results since it allows reviewers and readers to understand changes done during the development of the review, and authors to explain the needs for changes in case they have been made (Peters et al., 2015; Peters et al., 2020). Therefore, we believe that its publication separately from the final review is justified. 

Below, we respond in detail to the specific comments made for some of the sections of the protocol.

Abstract:

Line 33: Change ‘systematic review’ to just ‘review’ since you go on to propose a scoping review.

Done.

Line 50: Remove ‘systematic review’ from keywords since this is technically not a systematic review

Done.

Introduction:

While I think the rationale for why a scoping review is necessary is acceptable, the introduction provides no strong rationale as to why the literature needs a manuscript describing the methods of a proposed scoping review. What necessitates the need for knowing the methodological rigor (which would be described in the methods section of the scoping review anyways)? What is novel about your methodological approach to the scoping review? What gaps or limitations in the literature are these novel methods addressing? These are the types of questions that need to be addressed/discussed in the introduction for a protocol paper.

As for clinical trials, there is increasing evidence of the existence of publication bias for reviews (Silagy et al., 2002; Tricco et al., 2009). The existence of an a priori protocol helps increase the rigour and trustworthiness of these studies for several reasons: (1) it allows reviewers to plan carefully and thereby anticipate potential problems; (2) it allows reviewers to explicitly document what is planned before they start their review, enabling others to compare the protocol and the completed review to replicate review methods if desired, and to judge the validity of planned methods; (3) it prevents arbitrary decision making with respect to inclusion criteria and extraction of data; and (4) it may reduce duplication of efforts and enhance collaboration (Shamseer et al., 2015). Therefore, the development of a protocol is important in itself, as it pre-defines the objectives, methods, and reporting of the review and allows for transparency of the process, limiting the occurrence of reporting bias (Peters et al., 2015; Peters et al., 2020).

While the prospective registration of planned systematic reviews in some popular databases such as PROSPERO may contribute to making this a priori plan visible, scoping reviews are not eligible for registration in this database (https://www.crd.york.ac.uk/prospero/). The publication of their comprehensive protocols in scientific journals may be, therefore, a good option to overcome this issue, promoting the best practice (i.e., conduct of the review in accordance with a fully developed protocol, and reporting in line with the PRISMA guidelines) in the conduct and reporting of scoping reviews (The PLoS Medicine Editors, 2011). 

Taking all this into account, we have made some modifications in the introduction to specify that this is a protocol for conducting a scoping review, and the benefits derived from the publication of an a priori protocol have been discussed later in the discussion, as the reviewer suggested. However, we do not believe that it is necessary to extend the explanation of why a protocol is necessary when it is within the quality criteria for conducting a scoping review.

Line 72: phrase ‘allow sport and exercise science and medicine professionals’ is clunky, rephrase to improve readability.

Done.

Lines 78-90: Most of the information is redundant. Only need to discuss the low cost and accessibility of force plates one time.

This part has been modified as suggested:

These devices are becoming increasingly utilised in applied environments such as sports (20,21) due to the advent of affordable, commercially available force plate systems that have been validated against industry gold standard systems (22–24) and well-established criterion data analyses procedures (25). No longer, therefore, are most force plate tests being conducted via laboratory-grade systems located within a traditional research environment (e.g., University laboratories). In fact, millions of force plate tests are being conducted by practitioners each year, with this number likely to rise thanks to the quickness, portability, and valuable information that the modern force plate systems can provide practitioners without the requirement for additional technology, such as motion capture systems.

Lines 108-116: This paragraph is somewhat misleading as it implies that a scoping review was performed, not the proposal of a methodological framework for a potential scoping review.

This paragraph has been slightly modified as suggested:

Therefore, the existence of a review would help to improve practitioner decision-making around force plate test and variable selection in relation to injury prevention purposes. After a preliminary search, no published or in-progress scoping or systematic reviews was identified on this topic, so here we present a protocol for a scoping review where we will provide a descriptive overview of the currently utilised force plate methodologies with athletes. In this protocol, we pre-define the objectives, methods, and reporting of our upcoming scoping review.

Aim/Objectives: These are very ambitious and will result in a very large number of papers for the final analysis (even after screening papers out)

As tactical athletes are no longer a focal point in our review, the breadth of articles to be considered for inclusion will be reduced. Likewise, we think that the slight amendments we have done to the injury definition and inclusion criteria will also limit the number of papers to be included in the final analysis.

Methodology:

2.1.1 Participants: Why no age or level of performance exclusion criteria? You are including both sport and tactical populations, but excluding paralympic athletes due to ‘different injury profiles and demands of sports for this populations’; does that mean you consider injury profiles of traditional athletes and tactical athletes analogous? Do you think injury profiles, exposure and sports demands are similar between adolescent recreational athletes (e.g., high school varsity basketball) and even division I collegiate basketball players?

Following reviewers’ suggestions according to the scope of our review, we have limited the target population to sports athletes, so tactical populations will be excluded. Regarding sports athletes, we agree with the reviewer that those participating in organised sports (i.e., athletes involved in competitive events) and those doing recreational sports would probably present differences in injury profiles, sport demands and injury exposure levels, which will make them not comparable. Therefore, only athletes participating in organised sports will be considered. To consider a study sample as “recreational athletes” (and thus, exclude the study) the classification framework proposed by Mckay et al. (2022) will be used. We have specified this in the main text of the protocol and also in the Supplementary File 3. However, no additional restrictions (by age or level of performance) will be applied as we would like to provide an overview of the tests and metrics most typically used for each of these categories.

2.1.2 Concept: Line 158: are you only considering primary injuries or secondary musculoskeletal injuries as well such as osteoarthritis?

Thank you for this comment. We have clarified this in the text as follows:

No exclusion will be made based on the type of injury (e.g., primary and secondary injuries), as long as the study population is athletes (i.e., not former athletes) and, in those cases where the participants are injured, they are undergoing rehabilitation at the time of data collection. 

2.2 Sources: This is a very broad range of studies to include which will drastically increase the number of manuscripts you will be reviewing. Do you believe that quasi-experimental, case-control, and cross-sectional studies will provide valid evidence for a force plate and its associated test’s prognostic capability? I would argue it would not and will further muddy interpretations interfering with the objective to provide a use guide for practitioners.

Thank you for your comment. We believe that the inclusion of all types of study design can help us to have a better overview of the tests and metrics that have been proposed to assess the risk of injury and make decisions during the athlete's rehabilitation process. This will not only help us discuss whether the reporting of the tests and metrics among studies is adequate and reproducible, but also to show the quality of the published evidence in this regard by categorising the papers according to the OCEBM levels of evidence 2009.

2.3 Search strategy: If going to include tactical athlete populations would suggest searching DTIC as well since much of their work won’t be published in journals indexed in Medline.

As explained in previous responses, tactical athletes have been removed from the scope of our review.

Between what years is the search being conducted?

No restrictions will be applied in terms of data publication, so all studies published before the last update of the search strategy will be considered. The last update will be carried out once we have the first draft completed to identify papers recently published and thus, to maximise the currency of our review.

Line 199-201: Need to have a specific statements or questions for both levels of screening that is reported to show how you identified if study was included or excluded. Would suggest using at minimum three levels (i.e., title only screening, abstract and full text).

We will conduct all the study selection (and also data extraction) through Covidence (covidence.org), a software for managing systematic reviews. This software follows the last update of the PRISMA guidelines, which differentiates a two-step process for study selection: (1) title and abstract screening and (2) full-text assessment. This information has been added to the corresponding section of the protocol.

Discussion:

Line 233-236: This is true that it will increase transparency, but all this information would be presented in the actual review as well.

Please, see the reasons previously stated to publish an a priori protocol. We think it is a fundamental task to improve the quality of the evidence provided.

Line 236: Only two individuals screening for a review this size might take a long time, especially during the stages of data extraction.

In our research group, we have two individuals with previous experience in the development of systematics reviews and meta-analysis which will be working full-time on the proposed review. They will make important efforts to have the data extraction as soon as possible; however, we are aware of the potential delay from the beginning of the process to the completion of the first draft. This is the reason why we have planned to regularly update the search, with the last one done once we have the first version of the manuscript completed to maximise the currency of our work.

Line 240: I wouldn’t consider only reviewing peer-reviewed literature as a weakness for this review since there is also a lot of misinformation on force plate injury screening protocols (especially from commercial sector just trying to sell their force plate and software platform).

We understand what the reviewer says and we partially agree with that. But it is also true that, in social and health research, an important body of first-level grey literature (in terms of Adams et al. 2017) exists in practitioner journals, books and reports from public and private institutions. This first grey level contains literature produced by authors with high-expertise and published with a high degree of explicit and transparent criteria. Therefore, we consider that the exclusion of these non-peer-reviewed reports should be acknowledged among the limitations of our planned review.

References used in responses to Reviewer 2:

Adams RJ, Smart P, Huff AS. Shades of grey: guidelines for working with the grey literature in systematic reviews for management and organizational studies. International Journal of Management Reviews 2017;19:432–54

McKay AKA, Stellingwerff T, Smith ES, Martin DT, Mujika I, Goosey-Tolfrey VL, et al. Defining training and performance caliber: a participant classification framework. Int J Sports Physiol Perform. 2022;17(2):317–31.

Peters MDJ, Godfrey CM, Khalil H, McInerney P, Parker D, Soares CB. Guidance for conducting systematic scoping reviews. JBI Evid Implement. 2015;13(3):141–6. 

Peters MDJ, Godfrey CM, McInerney P, Munn Z, Tricco AC, Khalil H. Chapter 11: Scoping Reviews. In: Aromataris E, Munn Z, editors. JBI Manual for Evidence Synthesis. JBI; 2020.

Shamseer, L., Moher, D., Clarke, M., Ghersi, D., Liberati, A., Petticrew, M., ... & Stewart, L. A. Preferred reporting items for systematic review and meta-analysis protocols (PRISMA-P) 2015: elaboration and explanation. Bmj. 2015;349.

Silagy CA, Middleton P, Hopewell S. Publishing protocols of systematic reviews: Comparing what was done to what was planned. JAMA 2002;287:2831–2834.

The PLoS Medicine Editors. Best Practice in Systematic Reviews: The Importance of Protocols and Registration. PLoS Med 2011;8(2):e1001009. https://doi.org/10.1371/journal.pmed.1001009

Tricco AC, Pham B, Brehaut J, Tetroe J, Cappelli M. An international survey indicated that unpublished systematic reviews exist. J Clin Epidemiol 2009;62:617–623.

---

## [Decision Letter · Decision Letter 1]

21 Sep 2023

Force plate methodologies applied to the injury profiling and rehabilitation in sport: a scoping review protocol

PONE-D-23-07414R1

Dear Dr. McMahon,

We’re pleased to inform you that your manuscript has been judged scientifically suitable for publication and will be formally accepted for publication once it meets all outstanding technical requirements.

Kind regards,

Chris Connaboy

Academic Editor

PLOS ONE

Additional Editor Comments (optional):

Reviewers' comments:

Reviewer's Responses to Questions

**Comments to the Author**

1. Does the manuscript provide a valid rationale for the proposed study, with clearly identified and justified research questions?

Reviewer #1: Yes

Reviewer #2: Yes

2. Is the protocol technically sound and planned in a manner that will lead to a meaningful outcome and allow testing the stated hypotheses?

Reviewer #1: Yes

Reviewer #2: Yes

3. Is the methodology feasible and described in sufficient detail to allow the work to be replicable?

Reviewer #1: Yes

Reviewer #2: Yes

4. Have the authors described where all data underlying the findings will be made available when the study is complete?

Reviewer #1: Yes

Reviewer #2: Yes

5. Is the manuscript presented in an intelligible fashion and written in standard English?

Reviewer #1: Yes

Reviewer #2: Yes

6. Review Comments to the Author

You may also provide optional suggestions and comments to authors that they might find helpful in planning their study.

Reviewer #1: The authors have done a commendable job of addressing my concerns and the revised manuscript is much improved. I do not have further suggestions and believe it is ready for publication. Best of luck with the review.

Reviewer #2: The authors have addressed all my concerns and made substantial changes to improve the quality of the manuscript.

7. PLOS authors have the option to publish the peer review history of their article (what does this mean?). If published, this will include your full peer review and any attached files.

Reviewer #1: No

Reviewer #2: No

---

## [Editor Report · Acceptance letter]

29 Sep 2023

PONE-D-23-07414R1 

Force plate methodologies applied to injury profiling and rehabilitation in sport: a scoping review protocol 

Dear Dr. McMahon:

I'm pleased to inform you that your manuscript has been deemed suitable for publication in PLOS ONE. Congratulations! Your manuscript is now with our production department. 

Kind regards, 

on behalf of

Dr. Chris Connaboy 

Academic Editor

PLOS ONE